# Determining the Stir-Frying Degree of *Gardeniae Fructus* Praeparatus Based on Deep Learning and Transfer Learning

**DOI:** 10.3390/s22218091

**Published:** 2022-10-22

**Authors:** Yuzhen Zhang, Chongyang Wang, Yun Wang, Pengle Cheng

**Affiliations:** 1School of Technology, Beijing Forestry University, Beijing 100083, China; 2Institute of Chinese Materia Medica, China Academy of Chinese Medical Sciences, Beijing 100700, China

**Keywords:** *Gardeniae Fructus* Praeparatus, process degree, deep learning, transfer learning

## Abstract

Gardeniae Fructus (GF) is one of the most widely used traditional Chinese medicines (TCMs). Its processed product, *Gardeniae Fructus* Praeparatus (GFP), is often used as medicine; hence, there is an urgent need to determine the stir-frying degree of GFP. In this paper, we propose a deep learning method based on transfer learning to determine the stir-frying degree of GFP. We collected images of GFP samples with different stir-frying degrees and constructed a dataset containing 9224 images. Five neural networks were trained, including VGG16, GoogLeNet, Resnet34, MobileNetV2, and MobileNetV3. While the model weights from ImageNet were used as initial parameters of the network, fine-tuning was used for four neural networks other than MobileNetV3. In the training of MobileNetV3, both feature transfer and fine-tuning were adopted. The accuracy of all five models reached more than 95.82% in the test dataset, among which MobileNetV3 performed the best with an accuracy of 98.77%. In addition, the results also showed that fine-tuning was better than feature transfer in the training of MobileNetV3. Therefore, we conclude that deep learning can effectively recognize the stir-frying degree of GFP.

## 1. Introduction

Gardeniae Fructus (GF) is the dried ripe fruit of *Gardenia jasminoides* Ellis (Figure 1) of the Rubiaceae family. It has a long history of use, was originally recorded in *Shennong’s Herbal Classics*, and is found in many Asian countries, including China, Japan, Korea, Vietnam, India, and Pakistan [1]. GF is a dual-use resource for both medicine and food.

Stir-Frying is one of the traditional Chinese medicinal processing methods that meets the different requirements of treatment, dispensation, and preparation according to traditional Chinese medicine (TCM) theory. The purposes of stir-frying are to improve the efficacy of the raw drug or reduce its toxicity for clinical use [2,3]. As GF is bitter and cold, to weaken its bitterness and cold nature, the processed products of GF are often used in medicine, such as *Gardeniae Fructus* Praeparatus (GFP). These processed GF products are commonly used in treatments for purging fire, clearing heat, and stopping bleeding. With the development of modern science and technology and in-depth understanding of Chinese medicine, the processed products of GF are widely applied in Chinese medicine clinics. *The Chinese Pharmacopoeia* (Ch. P 2020) included the most common Chinese patent medicines containing Gardenia decoction pieces with 95 prescription preparations. These prescriptions include Zhiqin fever-clearing combination, Yin zhihuang oral solution, etc., and are widely applied in the market.

In the GFP-stir-frying process, it is important to identify the best stir-frying degree for GFP to determine the quality of the processed GFP. Among the different methods to identify stir-frying degrees, the change in appearance trait of GF is an important parameter in grading the ripeness of a gardenia concoction. At present, pharmacists often use changes in the surface and internal color to judge when to end the stir-frying process of GFP. *The Chinese Pharmacopoeia* (Ch. P 2020) stipulates that GFP should be stir-fried until the surface is burned brown or burned black and the inner surface of the fruit’s skin and the surface of the seed are yellow-brown or brown. However, experienced experts are inconsistent in their determination of the average stir-frying degree. As a result, the quality of GFP varies significantly between different manufacturers, and even between different batches of the same manufacturer. In the GFP-stir-frying process, there is a close relationship between safety and efficacy. The intermediate products produced in the stir-frying process may cause certain toxicity or side effects [2]. Therefore, standardizing the GFP-stirfrying process is of great importance.

In recent years, color analysis has become a widely used method to describe the appearance of Chinese herbal pieces, which can objectively turn the traditional experience of discriminating between appearances of the tablets into data. However, to facilitate the homogenization of the color of the Chinese herbal pieces, most researchers determine the color of the powder of the medicinal slices of TCM through the use of color analyzers to determine changes in color while stir-frying Codonopsis pilosula and GF [4,5]. Many researchers also use destructive methods, which are expensive, complex processes that do not meet the needs of real-time monitoring. In contrast, nondestructive and real-time monitoring is more reliable and cost-effective [6,7].

Although artificial intelligence is developing rapidly and is widely used in the fields of real-time and nondestructive monitoring, it has limited applications in monitoring the processing of Chinese herbs [8,9]. In the early days, researchers used traditional machine learning methods to artificially extract the color, texture, geometric characteristics, and other characteristics of agricultural products [10,11,12]. However, manual feature transfer has many limitations in the features that can be extracted due to its high complexity and long computation time. In contrast, deep learning can automatically identify and extract the best features during the learning process, saving time and labor. Researchers have explored the use of deep learning methods to identify the ripeness of agricultural products, such as jujube [13,14], passion fruit [15], grape [16], blueberry [17], and green pepper [18], and some have also built real-time grading devices. Furthermore, deep learning is more widely used in other tasks in agriculture [19], such as defect identification in agricultural products [20,21,22], disease classification [23,24], and object detection [25].

Nevertheless, training in deep learning requires a large amount of data support, and the lack of data may lead to local optimal solutions or overfitting [26]. Therefore, in this paper, we use transfer learning [27] to mitigate the lack of training data. Instead of training the network with random parameters, transfer learning first loads pretrained model parameters and then fine-tunes all the layers or freezes some layers and trains only the remaining parts. Through transfer learning, the model can be better adapted to the new dataset, thus increasing convergence and saving time [28,29,30,31]. 

Although deep learning may be a good candidate for monitoring the processing of Chinese herbs, it is yet to be studied. Therefore, the objective of this study is to establish a dataset of the different stir-frying degrees of GFP and apply deep learning models to its stir-frying degree recognition to quickly and accurately identify the appropriate stir-frying degree of GFP. In this study, five popular networks were selected and trained using the transfer learning mechanism, and the recognition results were compared and analyzed. The study provides a reference for the online monitoring technology of the stir-frying degree of GFP.

## 2. Materials and Methods

### 2.1. Data Acquisition

In this paper, the GF used for the analysis was purchased from Anguo City, Hebei Province, originating from Fengcheng, Jiangxi Province. The GF was identified as Gardenia jasminoides, a dried mature fruit of the Rubiaceae family, by Professor Zhang Cun, Institute of Chinese Materia Medica, Chinese Academy of Chinese Medical Sciences. *Gardeniae Fructus* Praeparatus (GFP) is the processed product of GF, which was processed based on *The Chinese Pharmacopoeia* (C. P. 2020), which specifies using the clear-stir-frying method (a stir-frying method without excipients). The stir-frying machine used, a CGY900B-type drum-type gas-frying machine (Jiangyin Xiangshan Traditional Chinese Medicine Machinery Co., Ltd., power 4 kW, power supply 380 V, frequency 50 Hz, pot diameter 90 cm, rpm 14 r·min^−1^), was produced by Beijing Ben Cao Fang Yuan Pharmaceutical Co (BBCFYPC).

The stir-frying process is described as follows: First, we crushed the raw GF according to the production standards of the BBCFYPC factory. When the stir-frying machine heated up to 260 °C, 10 kg of crushed raw GF was placed in the stir-frying machine at a medium heat, in accordance with the stir-frying without excipients method. Sampling was performed every 1 min with 100~200 g each time. The stir-frying process continued until the outer skin was scorched brown, the inner skin had scorched spots, and the seed was brownish. Then, the processed GF samples were taken out. An experienced expert identified the 12 min sample as the most moderately stir-fried sample. The above samples were for subsequent experiments.

The images of the GFP at various stir-frying degrees were collected. To ensure the consistency of the samples, every image was captured under the same conditions. We took pictures using a Xiaomi 10 s at a 1600 × 1200 pixel resolution and saved them in JPEG format. The stir-frying degree was divided into 8 stages. The change in the early stage of stir-frying was small, so the first 10 min were divided into stages of 2 min. As there was an obvious change in the later stage of stir-frying, which was close to the average stir-frying degree, the later sample was divided into a stage for every minute, including 1–2 min, 3–4 min, 5–6 min, 7–8 min, 9–10 min, 11 min, 12 min, and 13 min. These images were recorded as C1, C2, C3, C4, C5, C6, C7, and C8, respectively. Figure 2 shows the sample images of GFP under different stir-frying stages.

To obtain samples of the different stages, as well as simulate the state of GFP stir-frying as much as possible and acquire as many images as possible, we randomly disrupted the samples being photographed. In total, 16 images were taken for each minute of sample acquisition, with a total of 208 images acquired. Subsequently, each image was evenly divided and cropped into 9 images with 533 × 400 pixels, totaling 1872 images, as shown in Figure 3.

To further enrich the image dataset, simulate the actual situation, and ensure that the trained model has better robustness and generalizability, image augmentation was performed on the dataset before training [32,33]. The image augmentation methods used in this study include horizontal flip, vertical flip, horizontal–vertical flip, pretzel noise, luminance reduction by 0.6 times and luminance enhancement by 1.6 times. Accordingly, 8164 images were obtained, and the 8164 images were randomly divided into a training dataset, a validation dataset, and a test dataset in the ratio of 8:1:1. Figure 4 shows the data augmentation method for the image of the GFP being stir-fried, and Table 1 shows the results of data augmentation.

### 2.2. Neural Network Structure

In the field of deep learning, CNN is the most famous and commonly used algorithm. A convolutional neural network (CNN) is a feedforward neural network that has a deep structure and uses convolutional computation. A CNN can automatically learn the features of the input image. Alexnet [34] won the championship of ILSVRC in 2012, which started the deep learning boom in the field of computer vision. ResNet [35] proposed the residual module in 2015 and was a major advancement in the field of deep learning. As the number of network layers deepens and the number of network model parameters becomes larger, the need for lightweight networks has become greater. The artificially designed lightweight networks MobileNet series [36,37,38] and ShuffleNet series [39,40] are more popular at present. In this paper, five typical, commonly used neural networks in the history of CNN development were selected to train the task of determining the different frying degrees of GFP.

#### 2.2.1. VGG16

VGG16 [41,42] is a convolutional neural network model developed by the Visual Geometry Group (VGG) at the University of Oxford that was the winner of the 2014 ILSVRC object recognition algorithm. Vgg16 consists of 13 convolution layers, 3 full connection layers, and 5 pooling layers. Its outstanding feature is simple. It uses multiple 3 × 3 convolution kernel to replace the larger convolution kernel (11 × 11, 7 × 7, 5 × 5). The pooling layer adopts the same pool core parameters, and the pool core size is 2 × 2. The network structure is shown in Figure 5.

#### 2.2.2. GoogLeNet

GoogLeNet [43] is a new deep learning structure proposed by Szegedy et al. [44]. The innovation lies in its Inception structure, which expands the depth and width of the whole network. The structure diagram of the Inception module is shown in Figure 6, which consists of three convolutions (1 × 1, 3 × 3, 5 × 5) and one pooling layer, and a 1 × 1 convolution immediately before each convolution and after the pooling layer. This reduces the number of parameters while extracting richer features. The GoogLeNet network based on the Inception module contains three convolutional layers and nine Inception modules. The network structure of GoogLeNet is shown in Figure 7. 

#### 2.2.3. ResNet

To solve the problems of gradient disappearance and gradient explosion caused by the deepening of network layers, He et al. proposed the ResNet [35] residual network in 2015 [45]. The structure of the residual block is shown in Figure 8, where the input of the residual block is x and the output is F(x) + x.

F(x) is the residual function, and the model only needs to learn the residual function F and minimize the residual function F(x) to solve the network degradation problem and enhance the network performance. Different numbers of residual modules are stacked to form ResNet networks with different numbers of layers. Classical networks have 18 layers, 34 layers, 50 layers, 101 layers, etc. Given the complexity of this problem and the device environment, we chose ResNet-34 for training; the network structure of ResNet-34 is shown in Figure 9.

#### 2.2.4. MobileNetV2 and MobileNetV3

The MobileNet model [36,37,38] is a lightweight deep neural network proposed by Google for embedded devices such as cell phones [46]. The core idea is depthwise separable convolution. Depthwise separable convolution consists of depthwise (DW) and pointwise (PW) channels. DW is a channel that is convoluted by a convolution kernel, and the number of feature map channels generated by this process is the same as the number of input channels. PW is equivalent to an ordinary convolution operation, but the size of the kernel is 1 × 1 × M. M is the number of channels in the previous layer. PW combines the previous Feature Map in the depth direction to generate a new Feature Map, which can greatly reduce the amount of model computation and model parameters. The depthwise separable convolution is shown in Figure 10.

At the same time, MobileNetV2 proposes an inverse residual structure. First, the dimensionality is increased by 1 * 1 convolution, followed by the use of DW. Then, using 1 * 1 convolution, the dimensionality is reduced and the process of DW can extract more features. MobileNetV3 adds a lightweight attention mechanism, SENET [47], to the inverse residual structure, and the action mode of the attention mechanism is to adjust the weight of each channel and pool each channel of the obtained characteristic matrix. The inverse residual structure of MobileNetV3 is shown in Figure 11.

### 2.3. Training Strategies

This study used transfer learning. The CNNs used in the study were all pretrained on the ImageNet [48] dataset, and the last fully connected layer of the network was changed from 1000 to 8. For MobileNetV3, two different migration learning methods were tested in this study. The first, a feature transfer method, freezes the weights of all layers except the softmax classification layer. The frozen layers are equivalent to image feature extractors and only the linear softmax classifier needs to be trained. The second, a fine-tuning method, initializes the network using pretrained parameters and trains the entire network parameters by fine-tuning them with the new data to improve the performance of the network on the desired task. All other networks are not frozen and all the network parameters are fine-tuned based on the pretrained model parameters. The transfer learning approach, which is based on fine-tuning every layer of the network, is shown in Figure 12.

In terms of the parameter settings, the optimizer selects the computationally efficient Adam optimizer. The learning rate is set to 0.0001, the batch size is set to 16, and the number of iterations is 300. The experiments are conducted using the training and validation sets from the GFP dataset. After the model is trained, the model is tested using the test dataset, and the test results of the test dataset are analyzed as a measure of the model.

## 3. Results and Discussion

Figure 13 shows the validation accuracy (val_acc) and training loss (train_loss) values of MobileNetV3 trained with two different transfer learning methods, respectively. In Figure 13a, the horizontal coordinates indicate the epochs and the vertical coordinates indicate the train_loss values. The train_loss values decrease rapidly with an increase in the training epochs, followed by a slow decrease and eventual stabilization. In Figure 13b, the horizontal coordinate indicates the epochs and the vertical coordinate indicates the val_acc. In our study, we trained a total of 300 epochs. It can be seen that fine-tuning performs significantly better than feature transfer. The fine-tuning method stabilizes after 300 epochs, with a train_loss value of 0.057 and a val_acc value of 0.994, whereas the feature transfer train_loss value is still slowly decreasing and the val_acc value is slowly increasing, with a train_loss value of 0.691 and a val_acc value of 0.796. The convergence speed of fine-tuning method is also faster, while the feature transfer method needs more iterations to converge slowly. This may be accounted by the fact that the size of our dataset is small and has a low similarity to the pretrained dataset ImageNet. When we freeze the previous layers and train only the fully connected layers, it is difficult for the feature extraction layer of the pretrained model to learn the GF features from the image [49]. Thus, fine tuning all the layers achieves better performance.

Five currently popular classification networks were compared, and all models were trained on our dataset with 300 epochs. Figure 14 shows the val_acc and train_loss of the five networks trained. VGG16’s train_loss had several peaks during convergence, and its validation accuracy was even more volatile. In contrast, the other models slowly and steadily converged during training. Upon analysis, a possible reason for this is that no suitable optimizer or learning rate hyperparameters were chosen for VGG16, resulted in large fluctuations in model performance. In terms of training loss, it can be seen that of all the models, MobileNetV3 demonstrated the smallest loss, followed by ResNet34 and MobileNetV2. In terms of validation accuracy, MobileNetV3 is the most accurate, followed by MobileNetV2.

A comparison of the five models is listed in Table 2. The first five columns show the performance of the models in the test dataset, including accuracy, macro-averaged precision, macro-averaged recall, and macro-averaged F1-score. It can be seen that the accuracy of all models in the test dataset is above 95.82%. The most accurate is MobileNetV3 with 98.77%, followed by MobileNetV2, GoogLeNet, VGG16, and ResNet34 with 97.66%, 97.42%, 95.82%, and 97.29%, respectively. The last three columns show the number of parameters, memory occupied, and total model computation for the five models. It can be seen that MobileNetV3 is the most compact model and the smallest in all aspects. Therefore, MobileNetV3 demonstrated the best performance overall on this task.

We used a confusion matrix to further analyze the performance of MobileNetV3 on the test dataset. As shown in Figure 15, C1~C8 represent the eight categories representing each stage of stir-frying GFP. Each row represents the real category, and each column represents the predicted category. Each number in the table corresponds to the number of data points in the row category predicted as the column category, where the diagonal line is the number of correct predictions. It can be seen that two images of stage C6 are predicted as stage C7; one image of stage C7 is predicted as stage C5; one image of stage C8 is predicted as stage C1; one image of stage C1 is predicted as stage C2; two images of stage C2 are predicted as stage C1 and stage C3, respectively; and three images of stage C3 are predicted as stage C1. All other predictions are correct. 

Based on the confusion matrix, we introduce the evaluation metrics, R (recall) and P (precision):R = TP/(TP + FN) × 100%(1)
P = TP/(TP + FP) × 100%(2)

Here, TP, FP, and FN refer to “predicted positive, true positive”, “predicted positive, true negative”, and “predicted negative, true positive”, respectively. The accuracy and recall rates for the eight stages are shown in Table 3. It can be concluded that the C1, C2, and C3 stages are more likely to be confused with each other as the samples of these three stages are similar. The C5, C6, and C7 stages have some samples that may be confused with each other as the samples of these three stages are similar. The C8 stage has a picture that is predicted to be C1 stage; however, it could be an accidental outcome from the image.

## 4. Conclusions

In this study, a method of recognizing the stir-frying stage of GFP based on deep learning and transfer learning was proposed to determine the different stir-frying stages of GFP and provide technical support for the final implementation of the online monitoring of the stir-frying degree of GFP. First, the dataset of different stir-frying stages of GFP was constructed, and data augmentation was carried out on the original image to ensure the effectiveness of the model. GoogLeNet, VGG16, ResNet34, MobileNetV2, and MobileNetV3 were selected to train the GFP dataset based on their pretrained model parameters. In the training of MobileNetV3, both feature transfer and fine-tuning were adopted, and the training of other networks was fine-tuned for the whole network. The results show that MobileNetV3 fine-tuning outperforms only in training the fully connected layer, and the accuracy of all five models in the test dataset reaches more than 95.82%, among which MobileNetV3 performs the best and has the best performance. It has the smallest number of parameters and the highest accuracy rate of 98.77%. It can be seen that deep learning has unexpected results regarding the recognition of the stir-frying stage of GFP, especially the lightweight network MobileNetV3. In the future, we hope to further optimize the model and embed it into hardware equipment to finally perform the real-time online monitoring of the degree of GFP.

## Figures and Tables

**Figure 1 sensors-22-08091-f001:**
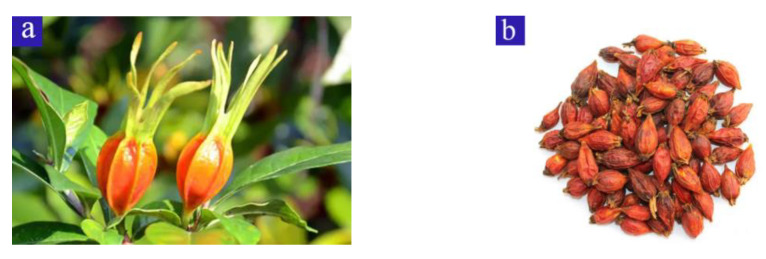
Gardeniae Fructus is the dried ripe fruit of *Gardenia jasminoides* Ellis. (**a**) Fresh fruit. (**b**) Gardeniae Fructus.

**Figure 2 sensors-22-08091-f002:**
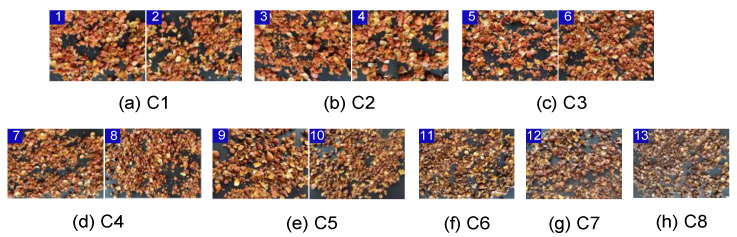
Images of GFP under eight different stir-frying stages. Note: C1 shows the process after 1 min, C2 shows the process after 2 min, and so on.

**Figure 3 sensors-22-08091-f003:**
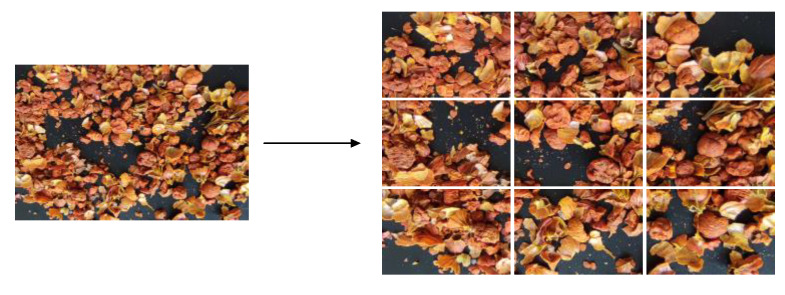
Example of image cropping process. The image on the left is the original, and the image on the right comprises the nine cropped images.

**Figure 4 sensors-22-08091-f004:**
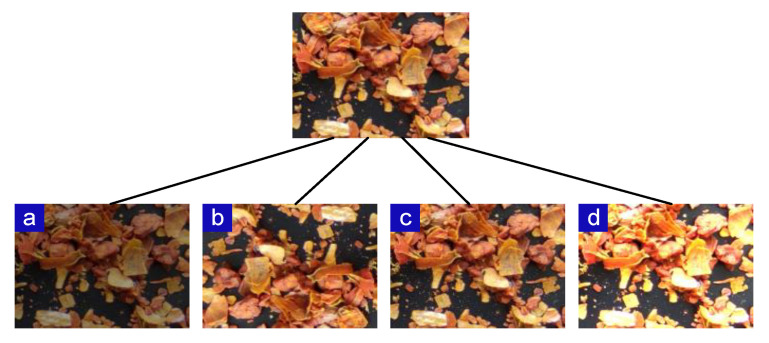
Examples of different augmented results. The isolated image above represents the original aspect. (**a**) Darkened image. (**b**) Overturned image. (**c**) Image after adding salt–pepper noise. (**d**) Brightened image.

**Figure 5 sensors-22-08091-f005:**
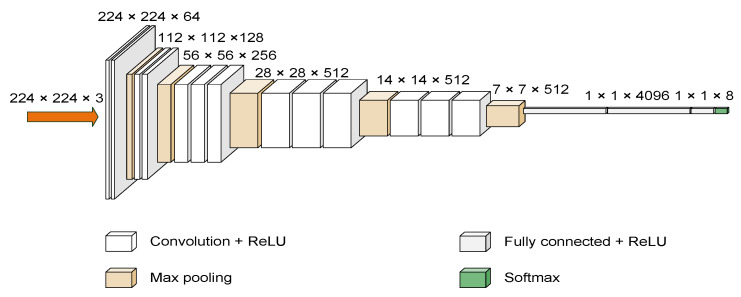
Image of VGG16 architecture with CNN.

**Figure 6 sensors-22-08091-f006:**
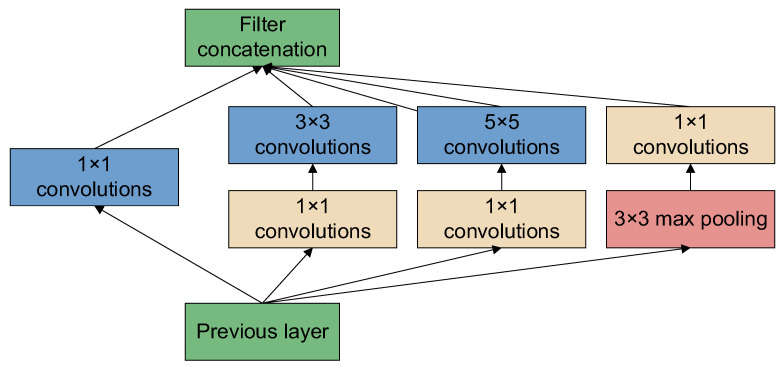
Structure diagram of the Inception module.

**Figure 7 sensors-22-08091-f007:**
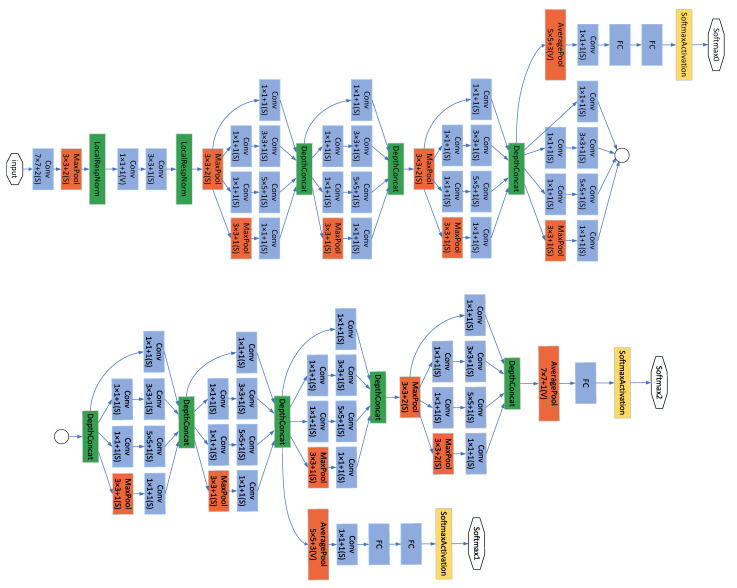
Image of GoogLeNet architecture. The circle connects the two parts of the figure.

**Figure 8 sensors-22-08091-f008:**
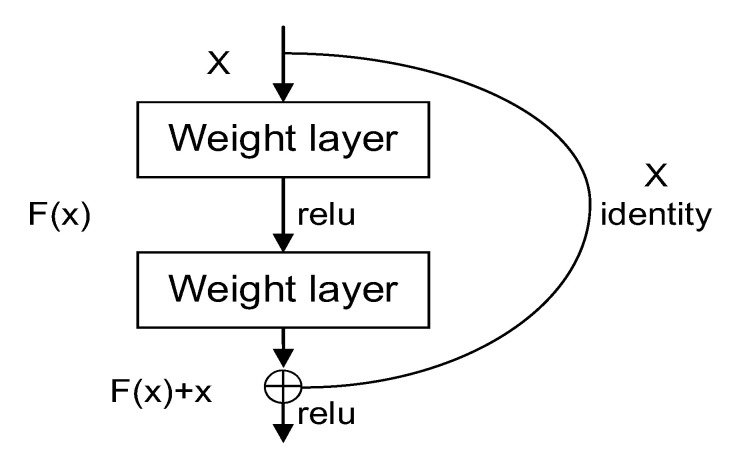
Residual neural architecture.

**Figure 9 sensors-22-08091-f009:**
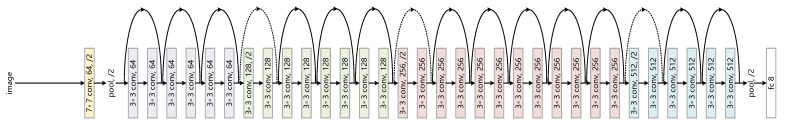
Image of ResNet-34 architecture.

**Figure 10 sensors-22-08091-f010:**
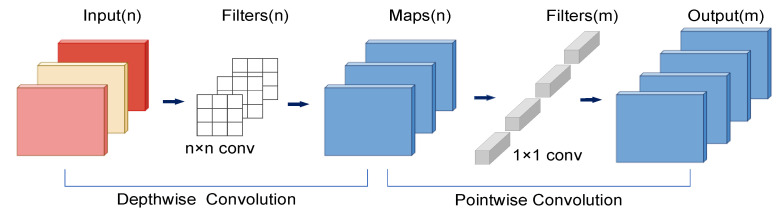
Depthwise separable convolution, including depthwise convolution (DW) and pointwise convolution (PW).

**Figure 11 sensors-22-08091-f011:**
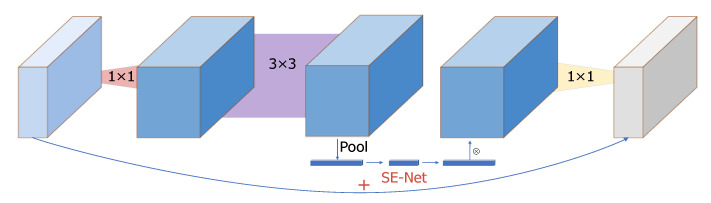
Inverse residual structure of MobileNetV3.

**Figure 12 sensors-22-08091-f012:**
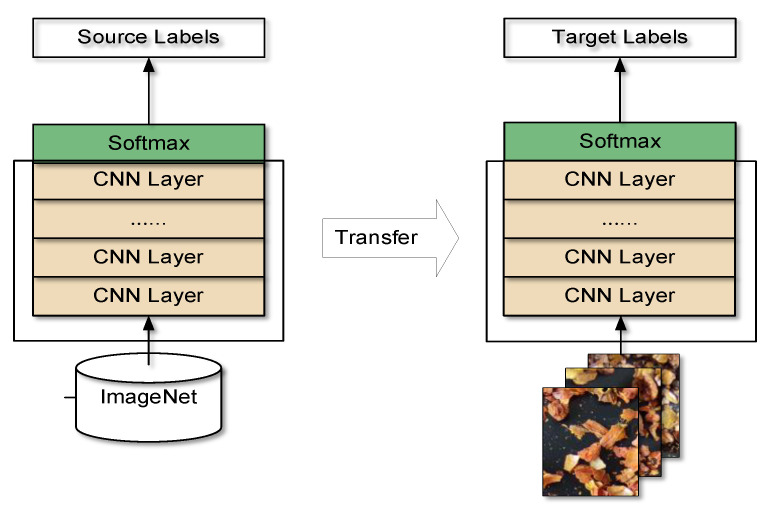
A transfer learning approach based on fine-tuning entire layers of the network to train a new dataset with weights that have been trained on ImageNet.

**Figure 13 sensors-22-08091-f013:**
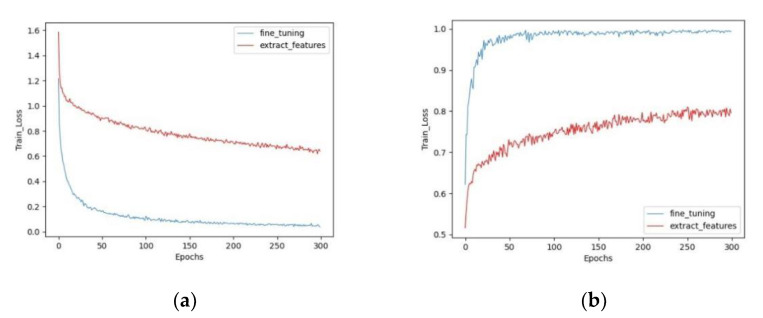
Accuracy and loss curve of MobileNetV3 in two different train methods, the blue line represents fine-tuning and the red line represents feature transfer. (**a**) Training loss (train_loss). (**b**) Validation accuracy (val_acc).

**Figure 14 sensors-22-08091-f014:**
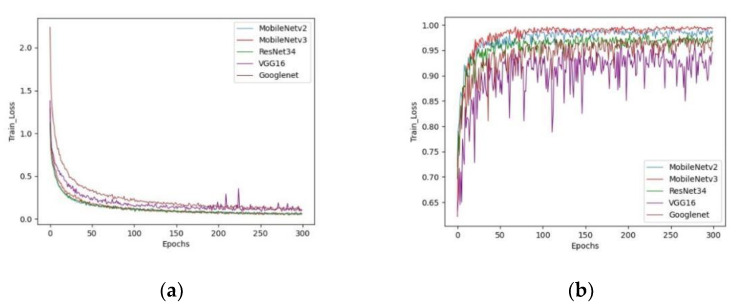
Accuracy and loss curve of different networks. (**a**) Training loss (train_loss). (**b**) Validation accuracy (val_acc). The train_loss of MobileNetV3 is the smallest and the val_acc is the highest.

**Figure 15 sensors-22-08091-f015:**
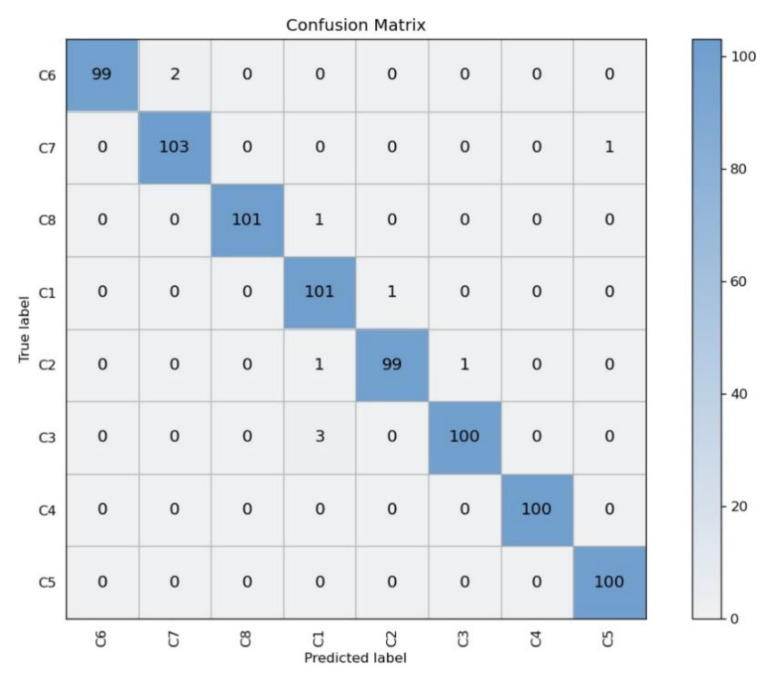
Confusion matrices of MobileNetV3 for the test image dataset.

**Table 1 sensors-22-08091-t001:** Number of images from the original database and expanded database.

Stage	Number	
	Original Database	Expanded Database
C1	288	1031
C2	288	1010
C3	288	1036
C4	288	1005
C5	288	1008
C6	144	1019
C7	144	1042
C8	144	1013
Total	1872	8164

**Table 2 sensors-22-08091-t002:** Comparison of the five models.

Models	Accuracy (%)	Precision (%)	Recall (%)	F1-score (%)	Params (Millions)	Model Size (MB)	Flops (MB)
MobileNetV3	98.77	98.80	98.78	98.78	1.53	16.19	58.79
MobileNetV2	97.66	97.73	97.67	97.68	2.23	74.25	318.97
ResNet34	97.29	97.32	97.29	97.30	21.29	37.61	36,700
VGG16	95.82	96.04	95.83	95.86	134.29	109.29	155,000
GoogLeNet	97.42	97.50	97.42	97.42	5.98	30.03	15,900

**Table 3 sensors-22-08091-t003:** The prediction result of the MobileNetV3.

	C1	C2	C3	C4	C5	C6	C7	C8
R (%)	99.02	98.02	97.09	100	100	98.02	99.04	99.02
P (%)	98.06	99.00	99.01	100	99.01	100	98.10	100

## Data Availability

Not applicable.

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
