# Peer review of "Determining the Stir-Frying Degree of Gardeniae Fructus Praeparatus Based on Deep Learning and Transfer Learning"

_sensors, 2022, doi:10.3390/s22218091_

Round 1
Reviewer 1 Report
Please revise your paper in the following parts.
1. Literature accumulation and listing are obvious in the introduction section, and further improvements are needed.
2. There are a lot of typos and grammar errors. It is better to conduct English proofreading.
3. Some references are missing in the paper, such as SENET (line 253) and Densenet-121 (line 103), YOLOv3 (line 86) ...
4. The technical novelty of this paper is not that significant.
5. The authors should detail the parameters of the image enhancement process.
Author Response
Dear reviewers:
We are very grateful to your comments for the manuscript. According with your advice, we tried our best to amend the relevant part and made some changes in the manuscript. All of your questions were answered below.
Once again, thank you very much for your comments and suggestions.
Yours Sincerely,
Yuzhen Zhang

Reviewer 2 Report
Comments from the reviewer:
This paper concentrates on the use of deep learning (DL) algorithms for the recognition of the frying degree of Gardeniae Fructus Praeparatus (GFP). The authors mentioned that 9224 images of GFP were prepared and assigned to 8 frying degrees (classes). Five DL models; VGG16, GoogLeNet, Resnet34, MobileNetv2, and MobileNetv3 were used to train the images based on the transfer learning strategy. The author reported that the MobileNetv3 outperformed other DL models for the training dataset. Based on the training performance, only MobileNetv3 was performed to quantify the recall rate and precision of the testing dataset. The paper was badly written and difficult to read. Many sentences were constructed too long which led the authors to use too many commas (,). A lot of issues need to be fixed before I can recommend it for publication.
Line 2-3: Please consider changing the Title: Recognition of processing degree for Gardeniae Fructus Praeparatus based on deep learning and transfer learning.
Line 12: “the intelligence of the processing of GFP is an urgent problem to be solved”. The problem in which aspect? For robot automation tasks or quality grading? Please specify according to your objective. Please also mention in the abstract that the scope of the processed GFP for this study is frying degree.
Line 17 -19: Why were two transfer learning strategies were applied to MobileNetv3 only? Why not apply the two strategies to all other DL models?
Line 19 -22: Please split the sentence properly.
Line 26-29: Please split the sentence properly.
Line 34: “Processing, is one of the traditional Chinese medicinal processing methods…..” should better replace “Frying, is one of the traditional Chinese medicinal processing methods…. ”
Line 40-46: Please split the sentence properly.
Line 47: replace processing degree with frying degree
Line 52: FPG or GFP?
Line 54-56: A lot of problems in the sentence lead to difficulty for the readers to understand. Please revise.
Line 64-66: which paper that you are referring to? Please state the references clearly.
Line 73: citation style issue. Please refer published paper in this journal on how in-text citation was constructed. Please check the same issue throughout this paper.
Line 74: what is BP?
Line 104: cnns should be upper case (CNNs)
Line 105: model apple or model for apple?
Line 119: which references said that?
Line 122: “fine-tunes the whole model” should replace with “fine-tunes the whole layers”
Line 132: Detection or recognition results?
Line 143-147: Did the GFP sample preparation follow the standard produced by Beijing Ben Cao Fang Yuan Pharmaceutical Co.? how many raw GF did you use (in grams or kg)? How did you arrange them inside the frying machine? What model of the frying machine did you use? Please describe what is clear frying method? What do you mean by the old medicine worker (is it referring to an experienced expert)?
Line 153: Why did you set the sample within a 2-minute frying interval for classes C1-C5 only? why not set all the classes with the same 2-minutes interval?
Line 164: What are the sizes of the cropped images?
Line 170 & 171: wrong terminology: you should state image augmentation not image enhancement.
Line 183-184: Table 1: According to line 162, if every minute you take 16 images and crop them into 9 images, the original database for C5 (9-10 min) should be 288. Table 1 only state 144. For the expanded database, why don’t you apply the augmentation process so that the number of the dataset is balanced? Let say for C1-C5 you augment with 4-folds, and C6-C9 with 8-folds. At the end of the augmentation, you will get a balanced amount of data.
Section 2.2.2 and Section 2.2.3. please also provide the illustration diagram of the model architecture for GoogLeNet and ResNet as you illustrate for VGG16 (Fig. 5) and MobileNet (Fig 8 & Fig 9).
Line 263 – 271: Two transfer learning strategies were applied on MobileNetV3. But the rest of the DL Neural Network models are only tuned using one strategy. Please state a clear reason why you did not standardize them.
Line 274: Please state the learning approach is based on fine-tuning entire layers of the network.
Line 288 – 293: Please discuss the reason why fine tuning is better than extracting features only.
Line 313: typo: val_acc not cal_acc
Line 322 – line 326: I suggest you perform the inference of the test dataset using all five models to further validate the training results depicted in Table 2. For all the models calculate the accuracy, precision, recall, and F1-score (present them in a table).
Line 328 & line 350 & 360 : typo: FPG or GFP?
Author Response

(The authors gave the same response as above.)

Round 2
Reviewer 2 Report
All of my previous comments were addressed properly. I wish to recommend this article for publication.